# Hippocampal neurogenesis enhancers promote forgetting of remote fear memory after hippocampal reactivation by retrieval

**Rie Ishikawa[1†], Hotaka Fukushima[1,2†], Paul W Frankland[3], Satoshi Kida[1,2*]**

[1]Department of Biosciences, Faculty of Applied Bioscience, Tokyo University of Agriculture, Setagaya-ku, Japan; [2]Core Research for Evolutionary Science and Technology (CREST), Japan Science and Technology Agency, Saitama, Japan; [3]Program in Neurosciences and Mental Health Program, The Hospital for Sick Children, Toronto, Canada

**Abstract** Forgetting of recent fear memory is promoted by treatment with memantine (MEM), which increases hippocampal neurogenesis. The approaches for treatment of post-traumatic stress disorder (PTSD) using rodent models have focused on the extinction and reconsolidation of recent, but not remote, memories. Here we show that, following prolonged re-exposure to the conditioning context, enhancers of hippocampal neurogenesis, including MEM, promote forgetting of remote contextual fear memory. However, these interventions are ineffective following shorter re-exposures. Importantly, we find that long, but not short re-exposures activate gene expression in the hippocampus and induce hippocampus-dependent reconsolidation of remote contextual fear memory. Furthermore, remote memory retrieval becomes hippocampus-dependent after the long-time recall, suggesting that remote fear memory returns to a hippocampus dependent state after the long-time recall, thereby allowing enhanced forgetting by increased hippocampal neurogenesis. Forgetting of traumatic memory may contribute to the development of PTSD treatment.

*For correspondence: kida@nodai.ac.jp

[†]These authors contributed equally to this work

**Competing interests:** The authors declare that no competing interests exist.

## Introduction

Post-traumatic stress disorder (PTSD) is a mental disorder associated with traumatic memory, including fear memory. In experimental animals PTSD may be modeled using Pavlovian fear conditioning. Pavlovian fear conditioning generates fear memory, reflecting an association between the conditioned stimulus (CS) and unconditioned stimulus (US). This CS-US association is stabilized and becomes a long-term memory (LTM) through gene expression dependent consolidation (*Flexner et al., 1965*; *Davis and Squire, 1984*; *Silva et al., 1998*; *McGaugh, 2000*; *Abel et al., 2001*).

Importantly, the strength of fear memories may be modified following their formation via several distinct processes. First, repeated and/or prolonged re-exposure to the CS induces extinction. This is an inhibitory learning process that reduces fear while preserving the original fear memory (*Pavlov, 1927*; *Rescorla, 2001*; *Myers and Davis, 2002*). Second, retrieval destabilizes a fear memory, necessitating protein synthesis restabilization (or reconsolidation). Accordingly, targeting protein synthesis following retrieval may prevent reconsolidation and weaken or even erase fear memories (*Nader et al., 2000*; *Debiec et al., 2002*; *Kida et al., 2002*; *Suzuki et al., 2004*). Third, forgetting processes may weaken established fear memories (*Akers et al., 2014*). While interventions to treat PTSD have focused on extinction and reconsolidation (*Bentz et al., 2010*; *Mueller et al., 2010*;

*Litz et al., 2012*), interventions that promote forgetting may represent an alternative approach for modifying traumatic memories.

In this regard, recently it has been shown that neurogenesis in the adult hippocampus regulates forgetting of hippocampus-dependent memories. Increasing hippocampal neurogenesis following training accelerated forgetting, whereas reducing hippocampal neurogenesis following training stabilized or strengthened existing hippocampus-dependent memories (*Akers et al., 2014*; *Epp et al., 2016*). Here we first replicate these findings and show that post-training elevation of hippocampal neurogenesis via memantine (MEM), which is a non-competitive inhibitor of the N-methyl-D-aspartate (NMDA) glutamate receptor (*Bormann, 1989*; *Seif el Nasr et al., 1990*; *Müller et al., 1995*; *Danysz et al., 1997*; *Volbracht et al., 2006*), can weaken contextual fear memories. As contextual fear memories age, they become less dependent upon the hippocampus for their expression (*Kim and Fanselow, 1992*; *Anagnostaras et al., 1999*; *Frankland and Bomtempi, 2005*; *Wiltgen et al., 2006*). Consistent with this, MEM treatment at remote time points failed to induce forgetting of contextual fear memories. However, reminders treatments that re-engage the hippocampus render contextual fear memories susceptible again to MEM-induced forgetting. As one of the major challenges in PTSD is finding treatments to weaken traumatic memories for events in the remote past although most approaches to weaken or erase fear memory target reconsolidation and extinction of the recent, but not remote, memory (*Nader et al., 2000*; *Monfils et al., 2009*; *Kaplan and Moore, 2011*; *Xue et al., 2012*), these findings suggest targeting forgetting processes as a potentially viable alternative or adjunct to extinction and reconsolidation based approaches.

## Results

### Forgetting of contextual fear memory by after MEM treatments

Post-training memantine (MEM) treatment was previously shown to enhance forgetting of contextual fear memory (*Akers et al., 2014*). We first confirmed these results under using our experimental conditions (*Figure 1A*). Mice were trained with a single foot shock (0.4 mA, Training), and 24 hrs later tested (Test 1). Twenty-four hrs after the Test 1, the mice received systemic injections of MEM (50 mg/kg body weight (bw)) or vehicle (VEH) once a week for four weeks (MEM-4 or VEH group). Another group received an injection of MEM only 24 hr after Test 1 (MEM-1 group). Contextual fear memory was assessed again four weeks after initial training (Test 2). All groups displayed comparable and high freezing response levels during Test 1. In contrast, the MEM-1 and -4 groups showed reduced freezing compared to the VEH group in Test 2 (*Figure 1A*), although this reduction was only statistically significant in the MEM-4 group. These observations were consistent with previous findings (*Akers et al., 2014*), and indicated that post-training MEM treatment enhanced forgetting in a dose-dependent manner.

Spontaneous recovery may occur after fear memory extinction (*Pavlov, 1927*; *Rescorla, 2004*; *Schiller et al., 2008*). To determine if spontaneous recovery was observed after MEM treatment, fear memory was assessed again at four weeks after Test 2 (Sp. Test). The MEM-4 group showed freezing that was comparable with the Test 2 group, and still showed significantly less freezing than the VEH group (*Figure 1A*). These observations suggested that spontaneous recovery does not occur following MEM treatment, supporting our conclusion that MEM enhances forgetting (rather than extinction).

### Correlation between forgetting and increased adult hippocampal neurogenesis

MEM treatments enhance forgetting through increases in adult hippocampal neurogenesis (*Maekawa et al., 2009*; *Akers et al., 2014*; *Ishikawa 2014*). Conversely, preventing MEM-associated increases in hippocampal neurogenesis blocks MEM-induced forgetting of contextual fear memory (*Akers et al., 2014*). We next examined the relationship between forgetting of contextual fear memory and increased hippocampal neurogenesis after MEM treatments. For this purpose, we examined the impact of different MEM doses on neurogenesis and forgetting. We performed a similar experiment to that shown in *Figure 1A* except mice were treated with low (25 mg/kg bw) or high (50 mg/kg bw) doses of MEM or VEH. Mice also received systemic injections of 5-bromo-2-deoxyuridine (BrdU; 50 mg/kg bw) to label proliferating cells, at two days after each MEM or VEH treatment,

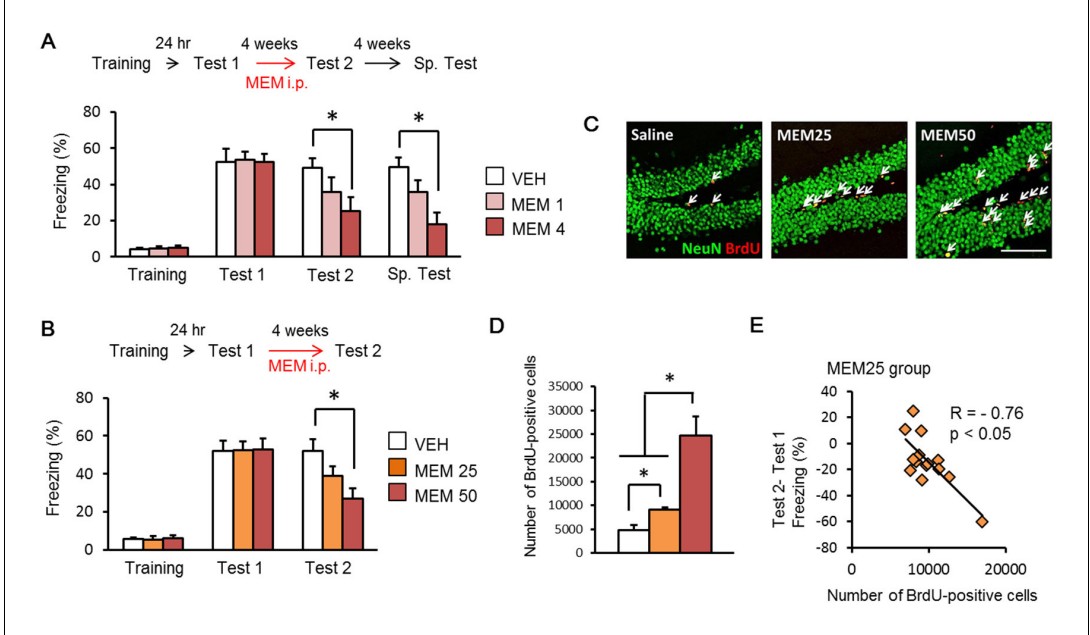

**Figure 1.** Memantine (MEM) treatment enhanced forgetting of contextual fear memory through the increase in adult hippocampal neurogenesis. (**A**) MEM treatment enhanced forgetting of contextual fear memory in a manner dependent on the number of MEM treatments. [VEH = vehicle-treated group, n = 10; MEM (1), n = 10; MEM (4), n = 10]. (**B**) MEM treatment enhanced forgetting of contextual fear memory in a dose-dependent manner (VEH group, n = 12; MEM25 group, n = 14; MEM50 group, n = 13). (**C**) Representative immunohistochemically stained BrdU-positive cells (red) and NeuN-positive cells (green) at 24 hr after Test 2. Scale bar = 100 μm. (**D**) The number of BrdU-positive cells in the dentate gyrus (DG) (VEH, n = 3; MEM25, n = 14; MEM50, n = 3). (**E**) Correlation between the number of BrdU-positive cells and the differences of freezing scores before and after the MEM treatments (MEM25 = 25 mg/kg body weight; MEM50 = 50 mg/kg body weight) in contextual fear conditioning tasks (n = 14). i.p. = intraperitoneal injection. *p<0.05. The results of the statistical analyses are presented in *Figure 1—source data 1*.

The following source data is available for figure 1:

**Source data 1.** Summary of statistical analyses with F values.

and then the number of BrdU-positive cells was quantified 24 hr after Test 2 by using immunohistochemistry. We found that MEM enhanced forgetting and hippocampal neurogenesis in a dose-dependent manner. The MEM-50 group showed significantly less freezing at Test 2, and significantly increased BrdU-positive cells compared to the VEH group (*Figure 1B*). The MEM-25 group showed intermediate levels of freezing and BrdU-positive cells compared to VEH and MEM-50 groups (*Figure 1C and D*). We then performed correlational analyses by comparing the number of BrdU-positive cells with reductions in freezing scores (by subtracting freezing levels of Test 1 from Test 2) for the MEM-25 group since this group that showed mid-levels of both forgetting and neurogenesis. We found that there was a significant negative correlation between forgetting and neurogenesis (*Figure 1E*). These results indicated that mice showing more neurogenesis also had more forgetting, and support the idea that MEM enhances forgetting through the increase in hippocampal neurogenesis.

## Forgetting after MEM treatment is dependent on hippocampus

In the adult brain neurogenesis persists in the subgranular zone (SGZ) of the hippocampus and in the subventricular zone (SVZ). Therefore, MEM-induced forgetting should be limited to brain regions like the hippocampus that are remodeled with the addition of new neurons (*Frankland et al., 2013*). To address this, we next trained mice in aversively-motivated tasks that are hippocampus-dependent and -independent, respectively. First, we used an inhibitory avoidance tasks that depends upon the hippocampus (*Lorenzini et al., 1996*; *Zhang et al., 2011*). Twenty-four hours after training with a 0.1 mA foot shock, mice were treated with MEM for four weeks (*Figure 2A*) and then crossover latency was assessed (Test). The MEM group showed significantly shorter crossover latencies than

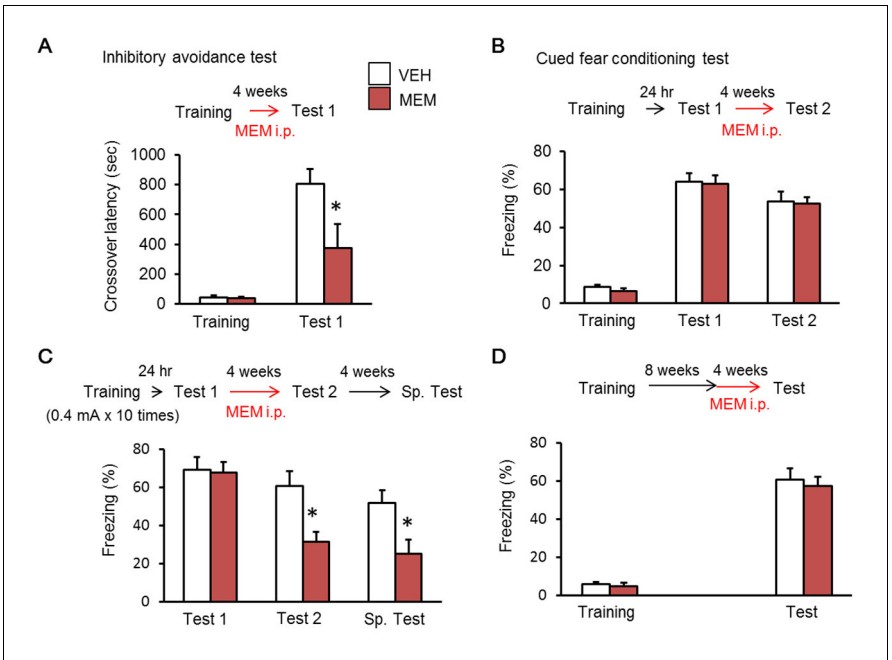

**Figure 2.** Memantine (MEM) treatment enhanced forgetting of hippocampus-dependent fear memory. (**A**) MEM treatment enhanced forgetting of hippocampus-dependent inhibitory avoidance memory [vehicle treated group (VEH), n = 11; MEM group, n = 10]. (**B**) MEM-treated mice showed normal amygdala-dependent memory (VEH, n = 12; MEM, n = 12). (**C**) MEM treatment affects strong fear memory (VEH, n = 8; MEM, n = 9). (**D**) Remote fear memory formed eight weeks after MEM treatment did not enhance forgetting (VEH, n = 10; MEM, n = 10). (**A**) and (**C**), *$p<0.05$, compared with the saline group. i.p. = intraperitoneal. The results of the statistical analyses are presented in *Figure 2—source data 1*.

The following source data is available for figure 2:

**Source data 1.** Summary of statistical analyses with F values.

the VEH group (*Figure 2A*), indicating that MEM enhanced forgetting of inhibitory avoidance memory. We next used cued fear conditioning tasks that can generate an amygdala-dependent fear memory (*Phillips and LeDoux, 1992*). Mice were presented a tone that co-terminated with a 0.4 mA foot shock in the training context (Training). Mice were assessed for their freezing responses to the tone in a different chamber at 24 hr (Test 1) and four weeks (Test 2) after the training. As in previous experiments, after Test 1 mice received MEM or VEH for four weeks. In contrast to the results shown in *Figure 1* and *Figure 2A*, the MEM group showed similar levels of freezing to the tone with the VEH group in both Test 1 and Test 2 (*Figure 2B*), indicating that MEM failed to promote forgetting of cued (amygdala-dependent) fear memory. These results are consistent with previous studies (*Akers et al., 2014*), and suggested that MEM promotes forgetting of only hippocampus-dependent fear memory.

We next examined the effect of MEM treatment on strong fear memory (*Figure 2C*). We performed similar experiments as shown in *Figure 1A*, except that mice were conditioned with 10 foot shocks. The MEM and VEH groups showed higher levels of freezing (over 60%) in Test 1 compared to the results shown in *Figure 1A*, suggesting that both groups formed strong fear memory. Interestingly, the MEM group showed significantly less freezing during Test 2 than the VEH group (*Figure 2C*), indicating the MEM enhances forgetting even when fear memory is strong. Importantly, the MEM group did not show spontaneous recovery during the Sp. Test, suggesting that MEM enhances forgetting of even strong fear memory.

The hippocampus-dependent memory becomes hippocampus-independent with time (*McClelland et al., 1995*; *Silva et al., 1998*; *Dudai, 2004*; *Wiltgen et al., 2004*; *Frankland and Bontempi, 2005*; *Squire and Bayley, 2007*). Therefore, we next asked whether MEM enhances

forgetting of remote contextual fear memory. Mice were trained, and then after a eight week delay, MEM treatment was initiated for four weeks. Following completion of this treatment freezing was assessed (Test). In contrast to the results shown in *Figure 1*, the MEM group showed comparable levels of freezing with the VEH group during this test. This result indicates that the MEM failed to enhance forgetting when memory was remote in time (*Figure 2D*).

## MEM treatment following prolonged context reminders enhances forgetting of remote memory

Our observation that MEM failed to promote forgetting of remote fear memory is perhaps not surprising because increases in neurogenesis should only impact memories that depend on the hippocampus. However, previously we showed that reminders (e.g., context re-exposure) may render memories hippocampus-dependent again and make them vulnerable again to amnestic treatments that block reconsolidation (e.g., protein synthesis blockade) (*Suzuki et al., 2004*). Using the same logic here we ask whether reminders (i.e., context re-exposure) may make remote contextual fear memories susceptible to MEM-induced forgetting. In our previous reconsolidation experiment we found that long (10 min) but not short (3 min) re-exposures to the context were effective, and so here we used these long and brief reminders. Mice were trained as previously, and then re-exposed to the context for 3 or 10 min eight weeks later (re-exposure). Following re-exposure, MEM treatment was initiated and memory tested four weeks later (*Figure 3A*). All groups showed comparable levels of freezing during the re-exposure (3 or 10 min). Four weeks later, the MEM group with 3 min re-exposure showed comparable freezing with the VEH group in the test. In contrast, the MEM group with 10 min re-exposure showed significantly less freezing than the other groups (*Figure 3A*). These results indicated re-exposure to the context for 10 min eight weeks after initial training renders the context memory sensitive to MEM-induced forgetting.

Exercise also promotes adult hippocampal neurogenesis, and induces forgetting of recent memory in a neurogenesis-dependent manner (*van Praag et al., 1999*; *Akers et al., 2014*). Therefore, we next examined the effects of exercise on forgetting of remote contextual fear memory. We performed a similar experiment as shown in *Figure 3A* except that the mice were given continuous access to a running wheel in their home cage (Run) or housed under sedentary conditions (No-Run) for four weeks after re-exposure to the context for 0, 3, or 10 min (re-exposure). Consistent with previous observations (*Figures 2* and *3A*), No-Run group and Run group without re-exposure did not weaken remote contextual fear memory. However, running for four weeks following long (10 min) but not brief (3 min) context re-exposure did lead to forgetting (*Figure 3B*). Together these experiments indicate that long (but not brief) reminders render remote contextual fear memories susceptible to forgetting induced by treatments associated with increased hippocampal neurogenesis.

## Reconsolidation of remote fear memory depends on the hippocampus

We next determined a possible mechanism by which extended pre-exposures render remote contextual fear memory sensitive to MEM-induced forgetting. Reconsolidation of contextual fear memory depends on the hippocampus (*Nader et al., 2000*; *Debiec et al., 2002*), and forgetting is associated with increased neurogenesis in the hippocampus. Therefore, we first evaluated whether extended context re-exposure make remote contextual fear memories freshly dependent on the hippocampus. To do this, we used a reconsolidation approach and asked whether a reminder renders a remote contextual fear memory sensitive to protein synthesis blockade. In this experiment, mice were trained and then re-exposed to the context 24 hr later for 3 min. Immediately following this re-exposure mice received a micro-infusion of the protein synthesis inhibitor anisomycin (ANI) or VEH into the dorsal hippocampus, and 24 hr later mice were assessed for freezing (Test). As we have previously observed (*Kida et al., 2002*; *Frankland et al., 2006*; *Suzuki et al., 2008*), ANI group exhibited reduced freezing during the subsequent test (*Figure 4A*), suggesting that protein synthesis is necessary for re-stabilization of fear memories following reminders. In a second experiment, mice were re-exposed to the context for 3 min or 10 min four weeks after training. In this case, ANI-infusions into the dorsal hippocampus impaired subsequent memory only following the longer reminder (*Figure 4B and C*). These results indicated that long reminders induce hippocampal gene expression-dependent reconsolidation of remote fear memory.

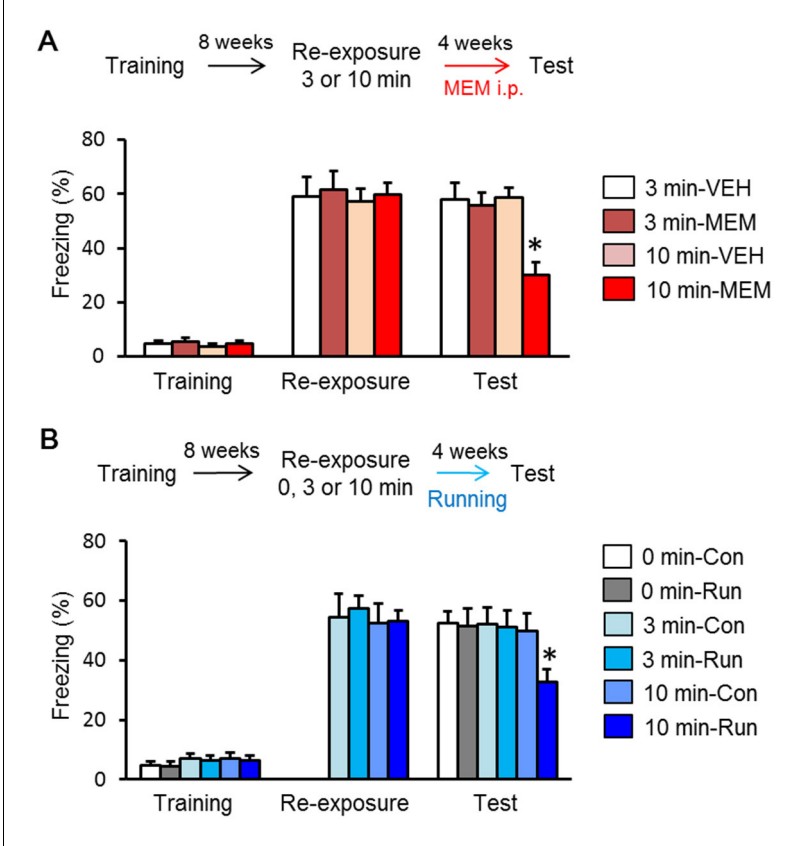

**Figure 3.** Enhancement of adult hippocampal neurogenesis after long-period recall enhanced forgetting of remote fear memory. (**A**) Memantine (MEM) treatment enhanced forgetting of remote fear memory after 10 min re-exposure (3 min vehicle treatment group (VEH), n = 10; 3 min MEM-treated group, n = 10; 10 min VEH, n = 12; 10 min MEM, n = 13). (**B**) Running after 10 min re-exposure enhanced forgetting of remote fear memory [No Run (Con), 0 min, n = 12; Run, 0 min, n = 9; No Run, 3 min, n = 12; Run, 3 min, n = 11; No Run, 10 min, n = 12; Run, 10 min, n = 11]. *p<0.05, compared with other groups in the test. The results of the statistical analyses are presented in *Figure 3—source data 1*.

The following source data is available for figure 3:

**Source data 1.** Summary of statistical analyses with F values.

## Long, but not short, context reminders reactivate the hippocampus at remote time points

Analysis of activity-regulated genes, including c-fos, indicates that retrieval of recently-acquired contextual fear conditioning memory is associated with activation of the hippocampus but not the anterior cingulate cortex (ACC). In contrast, retrieval of remote contextual fear memory induces *c-fos* expression in the ACC, but not in the hippocampus (*Frankland et al., 2004*). Therefore, the retrieval of remote memory appears not to reactivate the hippocampus. However, it is important to note that, in this previous study, contextual fear memory was retrieved by re-exposure to the context for only 3 min, which corresponds to our brief context reminder treatment used in the current study. It is possible that longer context reminders at remote time-points will induce c-fos expression in the hippocampus, a finding that would be consistent with our observation that longer context reminders render remote contextual fear memories sensitive to reconsolidation blockade by protein synthesis inhibition.

Therefore, we examined this possibility by measuring c-fos expression in the hippocampus and ACC following the retrieval of recent or remote memory, using immunohistochemistry (*Figure 5A*). The following four groups were used in this experiment: two groups which were trained with a foot

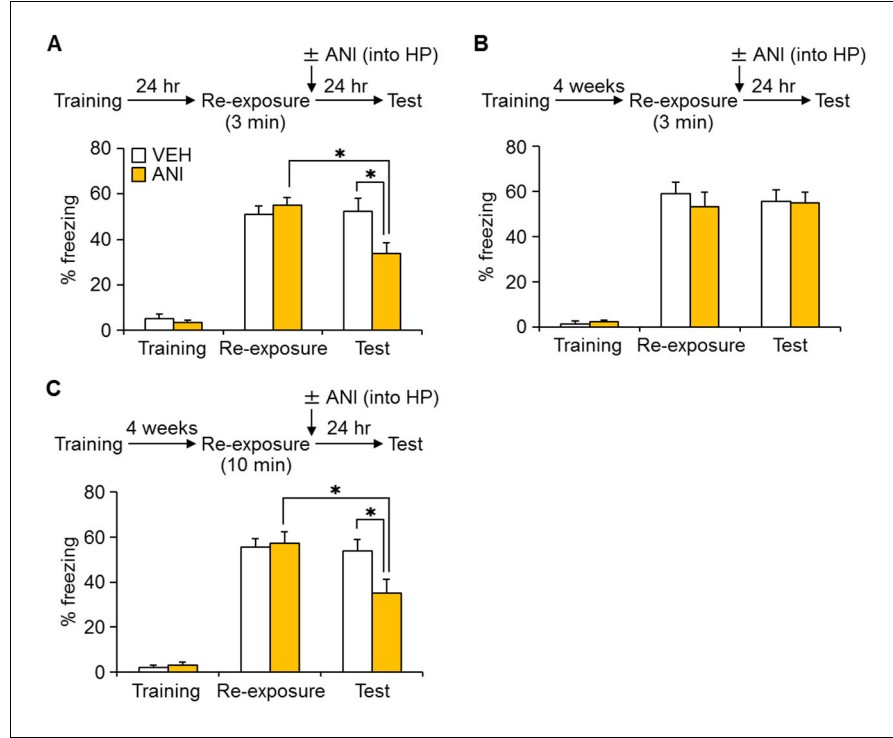

**Figure 4.** Inhibition of protein synthesis in the hippocampus (HP) blocked reconsolidation of remote fear memory. (**A**) Three min re-exposure to the context 24 hr after training [vehicle-treated group (VEH), n = 10; anisomycin (ANI), n = 15]. (**B**) Three min re-exposure to the context four weeks after training (VEH, n = 8; ANI, n = 8). (**C**) Ten min re-exposure to the context four weeks after training (VEH, n = 11; ANI, n = 11). Error bars, SEM. *p<0.05. The results of the statistical analyses are presented in *Figure 4—source data 1*.

The following source data and figure supplement are available for figure 4:

**Source data 1.** Summary of statistical analyses with F values.

**Figure supplement 1.** The cannula tip placement in the hippocampus.

shock (Trained), and the remaining two groups that did not receive a foot shock (Untrained). Twenty-four hr or four weeks after the training, mice were placed back in the training context for 3 or 10 min (re-exposure), and then assessed for c-fos expression 90 min after the re-exposure. In the ACC, both the brief and long re-exposures induced c-fos expression at the remote, but not recent, time point (*Figure 5B*). In contrast, in hippocampal CA1 and CA3 subfields, whereas brief re-exposure induced c-fos expression at the recent but not remote time points, longer re-exposures induced c-fos expression at both delays (*Figure 5B*). These observations support the idea that longer re-exposures are sufficient to re-engage the hippocampus, even at remote time-points to initial training.

## Retrieval of remote fear memory becomes dependent on the hippocampus following long context reminders

Our observations that long context re-exposures re-engage the hippocampus, even at remote time-points, suggest that remote contextual fear memory returns to a hippocampus-dependent state. To directly address this possibility, we examined the effects of hippocampal inactivation on the retrieval of remote memory after long-time retrieval. Mice were re-exposed to the context for 10 min, four weeks after the training (re-exposure). Twenty-four hr later, mice received a micro-infusion of the sodium channel blocker lidocaine (LIDO) or VEH into the hippocampus, and 10 min later, freezing was assessed (Test 1, *Figure 5C*). Mice were assessed for freezing again 24 hr after Test 1 (Test 2,

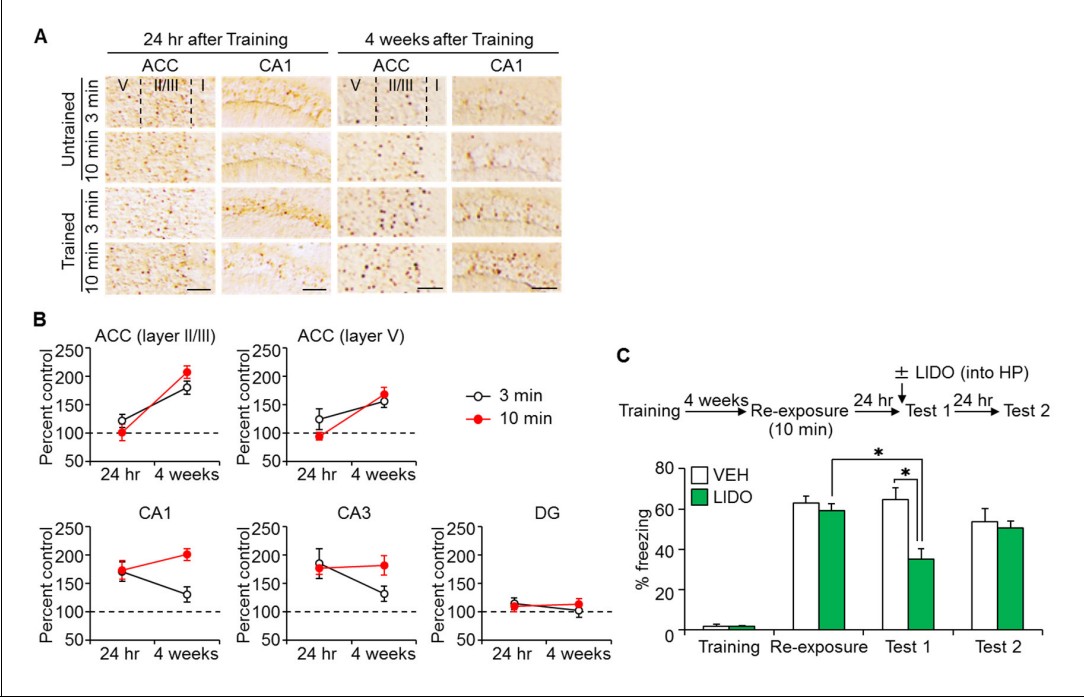

**Figure 5.** Long-time retrieval of remote fear memory reactivated the hippocampus (HP). (A) Representative immunohistochemical staining of the anterior cingulate cortex (ACC) and CA1 c-fos-positive cells from the indicated group. Scale bars = 100 μm. Two groups were trained with a foot shock (trained group) and the remaining two groups did not receive a foot shock (untrained group). Twenty-four hr (recent memory) or four weeks (remote memory) after the training, mice were placed back in the training context for 3 or 10 min for short or long memory recall, respectively (re-exposure). (B) The c-fos expression in the ACC (layers II/III, and V) and the hippocampus [CA1, CA3, and the dentate gyrus (DG) regions]. N = 10 for each group. The c-fos expression in the trained groups was expressed as a percentage relative to the untrained groups. (C) Ten min re-exposure to the context four weeks after training [vehicle-treated group (VEH), n = 11, lidocaine-treated group (LIDO), n = 11]. Error bars, SEM. *p<0.05. The results of the statistical analyses are presented in *Figure 5—source data 1*.

The following source data and figure supplement are available for figure 5:

**Source data 1.** Summary of statistical analyses with F values.

**Figure supplement 1.** The cannula tip placement in the hippocampus.

*Figure 5C*) in a drug-free state. LIDO-induced inactivation of the hippocampus impaired retrieval. Importantly, when retested drug-free, LIDO group exhibited comparable retrieval with VEH group (Test 2). These results suggest that the retrieval of remote memory becomes dependent on the hippocampus following the long re-exposure intervention, supporting our hypothesis that remote contextual fear memory returns to a hippocampus-dependent state with sufficient reminding.

## Discussion

In the current study, we build on recent studies showing that increasing hippocampal adult neurogenesis promotes forgetting (*Akers et al., 2014*; *Epp et al., 2016*). Here we used MEM treatment to enhance hippocampal neurogenesis (*Maekawa et al., 2009*; *Namba et al., 2009*; *Ishikawa et al., 2014*). We first demonstrated that MEM treatment promoted forgetting of 'recent' contextual fear memory that was hippocampus-dependent. We also showed that the enhanced forgetting induced by MEM correlated with increases in hippocampal neurogenesis. However, MEM failed to promote forgetting of amygdala-dependent fear memory. Furthermore, in contrast to recent memory, MEM had no impact on remote fear memory that was hippocampus-independent (*Dudai, 2004*; *Frankland and Bontempi, 2005*; *Squire and Bayley, 2007*). Collectively, our observations suggest

that increases in hippocampal neurogenesis promote forgetting of hippocampus-dependent recent memory, but not remote and/or hippocampus-independent memory.

In previous experiments we found that prolonged (10 min) context re-exposures may render even remote contextual fear memories labile and hippocampus-dependent (*Suzuki et al., 2004*). Accordingly, using the same approach here, we found that MEM treatment following 10 min re-exposure enhanced forgetting of remote contextual fear memory. In addition, forgetting of remote contextual fear memory was promoted by exercise using a running wheel, another hippocampal neurogenesis enhancer (*van Praag et al., 1999*; *Akers et al., 2014*) following long (but not short) context re-exposures. It is important to note that this 10 min re-exposure did not appear to trigger the extinction of remote contextual fear memory for a number of reasons. First, decreased freezing responses were observed only when mice were treated with hippocampal neurogenesis enhancers (MEM or running). This suggests that this decrease of freezing scores depends on hippocampal neurogenesis-induced forgetting rather than extinction. Second, reconsolidation and extinction of contextual fear memory require longer re-exposure when memory becomes older or stronger, suggesting that 10 min re-exposure to the context is sufficient for reconsolidation but insufficient for extinction when memory is remote (see *Figure 4*, *Suzuki et al 2004*, data not shown). Third, reconsolidation and extinction show contrasting biochemical signatures in the hippocampus (c-fos expression is observed in the hippocampus when memory is reconsolidated, but not extinguished) (*Mamiya et al., 2009*). Consistently, 10 min re-exposure induces hippocampal c-fos expression and enables disrupt fear memory by protein synthesis inhibition, suggesting that this longer re-exposure induces reconsolidation but not extinction. Taken together, our findings suggest that forgetting of remote contextual fear memory is enhanced by the increase in hippocampal neurogenesis only after extended behavioral 'reminders'.

We hypothesize that extended context re-exposures enable remote memory to return to a hippocampus-dependent state, thereby allowing enhanced forgetting by increases in hippocampal neurogenesis. Our findings support our hypothesis as follows: (1) Hippocampus-dependent reconsolidation of remote memory is induced following the long, but not short context re-exposures, suggesting that extended 'reminders' recruit the hippocampus to help regulate remote memory. (2) The long, but not short context re-exposure at remote time points induces the expression of c-fos in the hippocampus, indicating that the prolonged retrieval of remote memory reactivates the hippocampus at the molecular and cellular levels. (3) The retrieval of remote memory is hippocampus-dependent when the memory is recalled 24 hr after these reminder treatments. Additionally, these findings suggest that the activation of hippocampus by longer context re-exposures open a window both to induce hippocampus-dependent reconsolidation and hippocampal neurogenesis-dependent forgetting by returning remote memory to a hippocampus-dependent state.

Retrograde amnesia is caused by damage or inhibition of hippocampal function. Remote memory can be recalled, while recent memory cannot, under the conditions of retrograde amnesia (*Zora-Morgan and Squire, 1990*; *Kim and Fanselow, 1992*; *Anagnostaras et al., 1999*; *Manns et al., 2003*; *Squire et al., 2004*; *Frankland and Bontempi, 2005*; *Wiltgen et al., 2006*). In agreement with these studies, Frankland et al. showed that the expression of immediate early genes (*c-fos, Zif 268*) was induced when recent contextual fear memory was retrieved, while this expression was not observed when remote memory was retrieved, suggesting that the hippocampus is reactivated only when recent memory is retrieved (*Frankland et al., 2004*). However, our results indicated that the expression of c-fos in the hippocampus was induced only at remote time points following extended (but not brief) context re-exposure, in agreement with *Frankland et al. (2004)*. The hippocampus can, therefore, be reactivated following the retrieval of remote memory only when extended 'reminders' are given. In the future, it will be important to determine whether the same hippocampal neurons that are incorporated into the memory engram at the initial fear conditioning are reactivated when the hippocampus is reactivated by long-time retrieval of remote memory.

Reconsolidation is suggested to be a process not only to maintain memory, but also to enhance and/or update memory (*Nader et al., 2000*; *Dudai, 2002*; *Tronel et al., 2005*; *Fukushima et al., 2014*). Previously, our study showed that reconsolidation of recent contextual fear memory was induced following 3 min context re-exposures, while similarly brief reminders failed to induce the reconsolidation when memory was remote (*Suzuki et al., 2004*). However, longer context re-exposures (10 min) enabled the induction of reconsolidation of remote memory (*Suzuki et al., 2004*). Furthermore, we found that the reconsolidation of remote memory required reactivation of the

hippocampus, reflected by activated gene expression. Therefore, our observations suggest that reactivation of the hippocampus following extended 'reminders' may contribute to updating remote memory in order to add new information to the original remote memory. Additionally, the possibility still remains that the activation (but not reactivation) of hippocampus by longer re-exposure to the familiar context is required to generate new memory associated with this familiar context.

In this study, hippocampal neurogenesis enhancers (MEM and running) failed to completely abolish fear memories. This may be due to remaining fear memory that depends on brain regions other than hippocampus (*Figures 1* and *2*). However, previous studies have shown that similar residual levels of freezing responses are still observed when consolidation or reconsolidation of fear memory is disrupted by amnestic drugs or gene manipulations and when fear memory is extinguished (*Kida et al., 2002*; *Wei et al., 2002*; *Lee et al., 2004*; *Suzuki et al., 2004*; *Chen et al., 2005*; *Zhao et al., 2005*; *Lee et al., 2008*; *Suzuki et al., 2008*; *Mamiya et al., 2009*; *Inaba et al., 2015*). Therefore, we concluded that significant forgetting was observed following the treatments with neurogenesis enhancers. In addition, we found that c-fos expression was not induced in the ACC and hippocampus when c-fos expression was assessed after the retrieval test following the treatments with MEM for four weeks (data not shown), strongly suggesting that the MEM treatment induces forgetting of fear memory even though mice still showed low level of freezing.

Fear memory extinction is an inhibitory learning process used to reduce fear responses to fear memory retrieval. However, spontaneous recovery of fear responses is observed with time after fear memory is extinguished (*Pavlov, 1927*; *Rescorla, 2004*; *Schiller et al., 2008*). In this study, we showed that spontaneous recovery of the fear response is not observed over time following forgetting of contextual fear memory. This observation suggests that, in contrast to fear memory extinction, forgetting of fear memory through increases in neurogenesis permanently erases fear responses.

In conclusion, we showed that forgetting of remote contextual fear memory was promoted by hippocampal neurogenesis following long (but not short) context reminders. Our findings identify forgetting as a targetable process for the treatment of PTSD. In the future, it will be important to extend our findings by examining effects of increased hippocampal neurogenesis on pathophysiological alterations linked with a PTSD-like state. Furthermore, it will be important to extend our findings to target hippocampus-independent traumatic memory since traumatic memory associated with PTSD must be more complex than contextual fear memory used in this study as the rodent PTSD model (*Shin et al., 2006*; *Milad et al., 2007*; *Parsons and Ressler, 2013*; *VanElzakker et al., 2014*; *Desmedt et al., 2015*).

## Materials and methods

### Mice

All experiments were conducted according to the *Guide for the Care and Use of Laboratory Animals* of *the Japan Neuroscience Society* and the *Guide for the Tokyo University of Agriculture*. All the animal experiments were approved by the Animal Care and Use Committee of Tokyo University of Agriculture. Male C57BL/6N mice were obtained from Charles River (Yokohama, Japan). The mice were housed in cages of 5 or 6 mice per cage, maintained on a 12-h light/dark cycle, and allowed *ad libitum* access to food and water. The mice were at least eight weeks of age when tested. Testing was performed during the light phase of the cycle. All experiments were conducted blind to the treatment conditions of the mice.

### Drugs

MEM (Sigma-Aldrich, St. Louis, MO) was dissolved in 0.9% saline. The mice were injected intraperitoneally (i.p) with MEM at a dose of 25 or 50 mg/kg body weight once a week for four weeks, unless otherwise indicated. The control mice were injected with the same volume of 0.9% saline (vehicle, VEH). As shown in *Figure 1B*, two days after MEM injection, the mice received single injections with 50 mg/kg body weight of 5-bromo-2-deoxyuridine (BrdU; Sigma-Aldrich). The protein synthesis inhibitor anisomycin (ANI; Wako, Osaka, Japan) was dissolved in artificial cerebrospinal fluid (ACSF), and adjusted to pH 7.4 with NaOH. The sodium channel blocker lidocaine (LIDO, 4%; Sigma-Aldrich) was dissolved in phosphate-buffered saline (PBS) (*Frankland et al., 2004*).

## Behavioral experiments

Before the commencement of the behavioral experiments, the mice were handled individually for 3 min, each day for five days. Animal behavior was recorded using a video camera.

### Contextual fear conditioning task

The mice were trained and tested in conditioning chambers (17.5 × 17.5 × 15 cm) that had a stainless steel grid floor through which a foot shock could be delivered (*Suzuki et al., 2004*, *Fukushima et al., 2008*; *Suzuki et al., 2008*; *Mamiya et al., 2009*; *Suzuki et al., 2011*; *Inaba et al., 2015*). Training consisted of placing the mice in the chamber and delivering single electrical foot shock (2 s duration; 0.4 mA), 148 s later, and then the mice were returned to their home cage at 30 s after the foot shock (Training period). In *Figure 2A*, mice received 10 electrical foot shocks separated by 30 s intervals.

For the first experiment (*Figures 1* and *2*), memory was assessed at 24 hr (Test 1) and one week after the last MEM treatment (Test 2) by calculating the percentage of time spent freezing during 5 min when returned to the conditioning chamber. Freezing behavior (defined as complete lack of movement, except for respiration) was measured automatically by video (O'Hara & Co., Ltd., Tokyo, Japan) as described previously (*Anagnostaras et al., 2000*). In *Figures 1A* and *2A*, we assessed spontaneous recovery by the re-exposure of conditioned mice to the context for 5 min, four weeks after Test 2 (Sp. test).

For the second experiment (*Figure 3*), eight weeks after training, mice were re-exposed to the context for 3 or 10 min (re-exposure). Twenty-four hr after the re-exposure, mice started the MEM treatment for four weeks, and memory were assessed one week after the last MEM treatment. Percent freezing represents averages of freezing behavior during the re-exposure for 3 or 10 min, respectively.

For the third experiment (micro-infusion of ANI, *Figure 4*), the mice were trained as described above, and at 24 hr (recent memory) or four weeks (remote memory) later, they were placed back in the context for 3 or 10 min (re-exposure). The mice were micro-infused with ANI (62.5 μg) or VEH into the hippocampus immediately after re-exposure. Twenty-four hr after re-exposure the mice were once again placed in the context and freezing was assessed. Micro-infusions into the hippocampus (0.5 μL) were made at a rate of 0.25 μL/min. The injection cannula was left in place for 2 min after micro-infusion and the mice were then returned to their home cages. This dose of locally infused ANI inhibits > 90% of protein synthesis for at least 4 hr (*Rosenblum et al., 1993*).

For the fourth experiment (c-fos immunohistochemistry, *Figure 5*), the mice were divided into four groups: (1) Two groups of mice were trained as described above (Trained group), and at 24 hr (recent memory) or four weeks (remote memory) after the training, were re-exposed to the context for 3 or 10 min. The animals were then anesthetized with Nembutal (750 mg/kg, i.p) at 90 min after re-exposure; (2) Two groups received a training session in the absence of foot shock (Untrained group), and at 24 hr (recent memory) or four weeks (remote memory) after the training, mice were re-exposed to the context for 3 or 10 min. The animals were then anesthetized, as described above, at 90 min after reactivation.

For the fifth experiment (micro-infusion of lidocaine, *Figure 5*), the mice were trained as described above, and four weeks later, they were placed back in the context for 10 min (re-exposure). Twenty-four hr later, mice received a micro-infusion of lidocaine (20 μg, LIDO group) or vehicle (VEH group) into the hippocampus and 10 min later, freezing was assessed (Test 1). Mice were assessed for freezing again 24 hr after the Test 1 (Test 2). Micro-infusions into the hippocampus (0.5 μL) were performed as described above.

### Cued fear conditioning test

A similar procedure was used for cued fear conditioning as described previously (*Kida et al., 2002*). Mice were placed in the training context, and 2 min later, a 30 s tone (2800 Hz at 85 dB) co-terminated with a 2 s shock (0.75 mA). Mice were given a similar tone-shock pairing 1 min later and returned to the home cage 1 min after the final shock. Mice were placed in a different chamber for 2 min and the tone was replayed for 3 min 24 hr later (Test 1) and one week after the last MEM treatment (Test 2). Memory for cued (tone) fear conditioning was assessed as the percentage of time mice spent freezing during the 3-min tone.

## Inhibitory avoidance task

The inhibitory avoidance task was performed as previously described (*Suzuki et al., 2011*; *Zhang et al., 2011*; *Fukushima et al., 2014*). A step-through inhibitory avoidance apparatus (O'Hara, Ltd) consisted of a box with separate light (15.5 × 12.5 × 11.5 cm) and dark (15.5 × 12.5 × 11.5 cm) compartments. The light compartment was illuminated by a fluorescent light (2500 lux). During the training session, each mouse was allowed to habituate to the light compartment for 30 s, and the guillotine door was raised to allow access to the dark compartment. Latency to enter into the dark compartment was considered as a measure of acquisition. Immediately after the mouse entered the dark compartment, the guillotine door was closed. After 5 s, a foot shock (0.1 mA) was delivered for a total period of 2 s. Memory was assessed four weeks later as the crossover latency for the mouse to enter into the dark compartment when returned to the light compartment, as during the training period.

## Exercise

Twenty-four hr after re-exposure, mice in the running groups were given voluntary access to a single running wheel placed in their home cages for four weeks. Mice in the sedentary groups were similarly housed but not given running wheels.

## Immunohistochemistry

Immunohistochemistry was performed as described previously (*Mamiya et al., 2009*; *Suzuki et al., 2011*; *Zhang et al., 2011*; *Fukushima et al., 2014*; *Ishikawa et al., 2014*; *Inaba et al., 2015*). After anesthetization, all mice were perfused with 4% paraformaldehyde (for BrdU or the neuron-specific protein NeuN staining, containing 0.5% picric acid). The brains were then removed, fixed overnight, transferred to 30% sucrose, and stored at 4°C. Coronal sections (14 μm for BrdU/NeuN staining, and 30 μm for c-fos staining) were cut using a cryostat.

For BrdU/NeuN staining (*Ishikawa et al., 2014*), consecutive sections were boiled in citrate buffer solution containing 82 mM sodium citrate and 18 mM citric acid for 5 min and incubated with 2N HCl at 37°C for 30 min, followed by incubation in a blocking solution [Tris-buffered saline with Tween 20 (TBST) buffer plus 10% goat serum albumin]. The sections were incubated with a monoclonal rat anti-BrdU primary antibody (1:5000; Novus Biologicals, Littleton, CO, RRID:AB_10002608) and a monoclonal mouse anti-NeuN primary antibody (1:500; Millipore, Hayward, CA, RRID: AB_2298772) in the blocking solution overnight. Subsequently, the sections were washed with PBS and incubated for 2 hr with Alexa Fluor 594-conjugated goat anti-rat IgG (1:500; Invitrogen Life Sciences, Grand Island, NY, RRID: AB_2534075) and Alexa Fluor 488-conjugated goat anti-mouse IgG (1:500; Invitrogen, RRID: AB_2534069).

For c-fos staining (*Mamiya et al., 2009*; *Suzuki et al., 2011*; *Zhang et al., 2011*; *Fukushima et al., 2014*; *Inaba et al., 2015*), the sections were washed and preincubated in 3% $H_2O_2$ in methanol for 1 hr, followed by incubation in a blocking solution (PBS plus 1% goat serum albumin, 1 mg/mL bovine serum albumin, and 0.05% Triton X-100) for 3 hr. Consecutive sections were incubated with a polyclonal rabbit primary antibody for anti-c-Fos (Ab-5, 1:5000; Millipore, RRID: AB_2106755) in the blocking solution overnight. Subsequently, the sections were washed with PBS and incubated for 3 hr at room temperature with biotinylated goat anti-rabbit IgG (SAB-PO Kit; Nichirei Biosciences, Tokyo, Japan), followed by 1 hr at room temperature in the streptavidin-biotin-peroxidase complex (SAB-PO Kit).

## Quantification

Structures were defined anatomically according to the atlas of *Franklin and Paxinos (1997)*. All immunoreactive neurons were counted by an experimenter blind to the treatment conditions.

For quantification of BrdU-positive cells, all fluorescence images were acquired using a confocal microscope (TCS SP8; Leica, Wetzlar, Germany). Equal cutoff thresholds were applied to all slices using LAS X software (Leica). BrdU-positive cells throughout the rostro-caudal extent of the DG were counted in every eighth section, and the total number of BrdU-positive cells was calculated by multiplying the count in each section by 8 and then totaling the values (*Ishikawa et al., 2014*). BrdU-positive cells were colocalized with NeuN, a marker of mature neurons, and were counted.

For quantification of c-Fos-positive cells in sections (100 × 100 μm) of the dorsal hippocampus (bregma between −1.46 and −1.82 mm) and anterior cingulate cortex (ACC, bregma between 0.8 and 1.0 mm) computerized image analyses were as described previously (WinROOF version 5.6 software; Mitani Corporation, Fukui, Japan) (*Frankland et al., 2006*; *Suzuki et al., 2008*; *Mamiya et al., 2009*; *Suzuki et al., 2011*; *Zhang et al., 2011*; *Fukushima et al., 2014*; *Inaba et al., 2015*). Immunoreactive cells were counted bilaterally with a fixed sample window across at least three sections. The expression levels of c-fos in the trained groups were expressed as a percentage relative to the untrained groups.

## Surgery for drug micro-infusions

Surgery was performed as described previously (*Frankland et al., 2006*; *Suzuki et al., 2008*; *Mamiya et al., 2009*; *Kim et al., 2011*; *Suzuki et al., 2011*; *Zhang et al., 2011*; *Nomoto et al., 2012*; *Fukushima et al., 2014*; *Inaba et al., 2015*). Under Nembutal anesthesia, and using standard stereotaxic procedures, a stainless steel guide cannula (22 gauge) was implanted into the dorsal hippocampus (−1.8 mm, ± 1.8 mm, −1.9 mm). Stereotaxic coordinates for dorsal hippocampus placement were based on the brain atlas of *Franklin and Paxinos (1997)*. The mice were allowed to recover for at least one week after surgery. After that, they were handled for one week before the commencement of inhibitory avoidance. Only mice with a cannulation tip within the boundaries of the dorsal hippocampus were included in the data analyses. Cannulation tip placements are shown in *Figure 4—figure supplement 1* and *Figure 5—figure supplement 1*.

## Data analyses

Data were analyzed using analysis of variance (ANOVA). One-way or two-way ANOVA followed by a *post hoc* Newman-Keuls or Fisher PLSD (*Figure 3B*) comparison were used to analyze the effects of groups, times, and drugs. A Student *t*-test was used to analyze differences of the freezing levels, crossover latency, and the number of BrdU-positive cells within each group. Pearson's correlation test was used to analyze the relationship between the number of BrdU-positive cells and freezing levels. All values in the text and figures represent the mean ± standard error of the mean. Statistical analysis was performed using StatView version 5.0 (SAS Institute Inc., Cary, NC).

## Acknowledgements

SK was supported by Grant-in-Aids for Scientific Research (A) (15H02488), Scientific Research (B) (23300120 and 20380078), and Challenging Exploratory Research (24650172 and 26640014), Grant-in-Aids for Scientific Research on Priority Areas - Molecular Brain Science- (18022038 and 22022039), Grant-in-Aid for Scientific Research on Innovative Areas (Research in a proposed research area) (24116008, 24116001, and 23115716), Core Research for Evolutional Science and Technology (CREST), Japan, The Sumitomo Foundation, Japan, and the Takeda Science Foundation, Japan.

## Additional information

### Funding

| Funder | Grant reference number | Author |
| --- | --- | --- |
| Japan Society for the Promotion of Science | Grant-in-Aids for Scientific Research 15H02488 | Satoshi Kida |
| Japan Society for the Promotion of Science | Grant-in-Aids for Scientific Research 23300120 | Satoshi Kida |
| Japan Society for the Promotion of Science | Grant-in-Aids for Scientific Research 20380078 | Satoshi Kida |
| Japan Society for the Promotion of Science | Challenging Exploratory Research 24650172 | Satoshi Kida |
| Japan Society for the Promotion of Science | Challenging Exploratory Research 26640014 | Satoshi Kida |

| | | |
|---|---|---|
| Japan Society for the Promotion of Science | Grant-in-Aids for Scientific Research on Priority Area - Molecular Brain Science 18022038 | Satoshi Kida |
| Japan Society for the Promotion of Science | Grant-in-Aids for Scientific Research on Priority Area - Molecular Brain Science 22022039 | Satoshi Kida |
| Japan Society for the Promotion of Science | Grant-in-Aid for Scientific Research on Innovative Areas 24116008 | Satoshi Kida |
| Japan Society for the Promotion of Science | Grant-in-Aid for Scientific Research on Innovative Areas 24116001 | Satoshi Kida |
| Japan Society for the Promotion of Science | Grant-in-Aid for Scientific Research on Innovative Areas 23115716 | Satoshi Kida |
| Core Research for Evolutional Science and Technology | | Satoshi Kida |
| Sumitomo Foundation | | Satoshi Kida |
| Takeda Science Foundation | | Satoshi Kida |

The funders had no role in study design, data collection and interpretation, or the decision to submit the work for publication.

## Author contributions

RI, HF, Co-wrote the manuscript, Performed behavioral and immunohistochemical analyses, Acquisition of data, Analysis and interpretation of data, Contributed unpublished essential data or reagents; PWF, Supervised experimental analyses, Analysis and interpretation of data; SK, Co-wrote the manuscript, Responsible for the hypothesis development and overall design of the research and experiments, Supervised the experimental analyses, Conception and design, Acquisition of data, Analysis and interpretation of data, Drafting or revising the article, Contributed unpublished essential data or reagents

## Author ORCIDs

Paul W Frankland, http://orcid.org/0000-0002-1395-3586
Satoshi Kida, http://orcid.org/0000-0002-8038-9583

## Ethics

Animal experimentation: All experiments were conducted according to the Guide for the Care and Use of Laboratory Animals (Japan Neuroscience Society) and the Guide for the Tokyo University of Agriculture. All the animal experiments were approved by the Animal Care and Use Committee of Tokyo University of Agriculture (authorization number: 250003). All surgical procedures were performed under Nembutal anesthesia, and every effort was made to minimize suffering.

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
