## [Decision Letter]

Thank you for submitting your article "Hippocampal neurogenesis enhancers promote forgetting of remote fear memory after hippocampal reactivation by retrieval" for consideration by *eLife*. Your article has been favorably evaluated by Sabine Kastner as the Senior Editor and three reviewers, including Karim Nader (Reviewer #2) and a member of our Board of Reviewing Editors.

The reviewers have discussed the reviews with one another and the Reviewing Editor has drafted this decision to help you prepare a revised submission.

Summary:

This is an intriguing study on the use of neurogenesis-promoting interventions to promote forgetting of remote fear contextual memories, like those implicated in PTSD, by re-engaging the hippocampus through prolonged exposure to the CS. In general, this is an elegant study that confirms the role of the adult hippocampus in forgetting, and suggests that this approach can be used to target remote fear memories.

Essential revisions:

The reviewers raised some concerns regarding the interpretation of some of the results as follows:

1) A constant issue in the fear conditioning literature is that extinction and forgetting/erasure have the same output (lack of freezing). In Mamiya et al., J. Neuroscience 2009 29:402, the same group submitting this paper showed that longer exposures to context (in this case 30 min instead of 10 min) activated extinction learning. While in Figure 1 and Figure 2 second test four weeks later showed that the decrease in freezing caused by increased neurogenesis did not spontaneously recover (as might be expected for extinction), this control was not done in Figure 3, where extinction was more likely due to the increased exposure to the context. Some explanation of this is needed, and some discussion of whether the difference between the 10 minute and 3 minute exposures may have some contributions from increased extinction due to the increased exposure to context.

2) The increase in c-FOS seen in the hippocampus after longer re-exposure is interpreted as enabling "remote memory to return to a hippocampus-dependent state". However, an alternative explanation could be that longer exposure to a previously familiar context generates a new memory, as opposed to reactivation of an old memory. This should be discussed. This may be what the authors are implying by pointing out the importance of whether the same hippocampal neurons are activated during the initial memory and the reactivation of the hippocampal memory by a prolonged exposure.

3) There is a concern that rodent PTSD models based solely on fear conditioning do not distinguish between a physiological response to a stressful event and the pathophysiological alterations that may more closely reflects a PTSD-like state (Desmedt et al. al Biol Psych 2015). Furthermore, it appears that neurogenesis-promoting treatments would only be effective for hippocampal dependent memories (Figure 2). In this regard, PSTD memories are probably much more complex than those acquired during contextual-fear conditioning.

4) The treatments (memantine and voluntary exercise) do not completely abolish the fear memories (recent or remote), reducing conditioned freezing from ~50-60% to ~ 20-30% (the latter appears to be greater than the baseline/unconditioned freezing in most experiments). There should be some discussion of what this remaining memory represents, and whether it is due to remaining fear memory, sensitization, or non-hippocampal representations of fear.

---

## [Author Response]

*Essential revisions:*

*1) A constant issue in the fear conditioning literature is that extinction and forgetting/erasure have the same output (lack of freezing). In Mamiya et al., J. Neuroscience 2009 29:402, the same group submitting this paper showed that longer exposures to context (in this case 30 min instead of 10 min) activated extinction learning. While in Figure 1 and Figure 2 second test four weeks later showed that the decrease in freezing caused by increased neurogenesis did not spontaneously recover (as might be expected for extinction), this control was not done in Figure 3, where extinction was more likely due to the increased exposure to the context. Some explanation of this is needed, and some discussion of whether the difference between the 10 minute and 3 minute exposures may have some contributions from increased extinction due to the increased exposure to context.*

Thank you for your critical comment. In Figure 3, we observed decreased freezing responses at the Test 4 weeks after the re-exposure to the context for 10 min only when mice were treated with MEM or running. In contrast, untreated mice did not show reduced freezing responses, suggesting that this decrease of freezing scores depends on increased hippocampal neurogenesis, but not due to memory extinction (Figure 1, Figure 2 and Figure 3). In addition, our previous studies have suggested that the induction of reconsolidation and extinction requires longer duration of re-exposure to the context when memory becomes older or stronger, suggesting that 10 min re-exposure to the context is sufficient to induce reconsolidation but fails to induce extinction when memory is remote (Suzuki et al. 2004, data not shown). Consistently, reconsolidation, but not extinction, was observed following 10 min re-exposure when memory is remote (Figure 4). Furthermore, our previous study (Mamiya et al., 2009) showed that c-fos expression is observed in the hippocampus when memory is reconsolidated in response to the context for 3 min, while this c-fos expression is not observed when memory is extinguished in response to the context for 30 min, indicating that reconsolidation and extinction shows contrasting biochemical signatures in the hippocampus. Thus, c-fos expression in the hippocampus is thought to be a biochemical marker of reconsolidation. Similarly, in this manuscript, c-fos expression was observed when remote memory is reconsolidated in response to 10 min re-exposure, suggesting that this longer re-exposure induces reconsolidation but not extinction. We believe that these 3 lines of evidence suggest that longer re-exposure to the context (for 10min) does not contribute to memory extinction, but helps to make the memory hippocampus-dependent again and therefore freshly vulnerable to the effects of neurogenesis-induced forgetting. According to the suggestions from review, we revised text in the Discussion (second paragraph).

“In previous experiments we found that prolonged (10 min) context re-exposures may render even remote contextual fear memories labile and hippocampus-dependent (Suzuki et al., 2004). […] Taken together, our findings suggest that forgetting of remote contextual fear memory is enhanced by the increase in hippocampal neurogenesis only after extended behavioral “reminders”.”

*2) The increase in c-FOS seen in the hippocampus after longer re-exposure is interpreted as enabling "remote memory to return to a hippocampus-dependent state". However, an alternative explanation could be that longer exposure to a previously familiar context generates a new memory, as opposed to reactivation of an old memory. This should be discussed. This may be what the authors are implying by pointing out the importance of whether the same hippocampal neurons are activated during the initial memory and the reactivation of the hippocampal memory by a prolonged exposure.*

Thank you for your important comment. We agree with the suggestion from the reviewer. According to this suggestion, we now mention the possibility raised by the reviewer (see below).

In addition, we have already mentioned to importance to examine “whether the same hippocampal neurons are activated during the initial memory and the reactivation of the hippocampal memory by a prolonged exposure”, as “In the future, it will be important to determine whether the same hippocampal neurons that are incorporated into the memory engram at the initial fear conditioning are reactivated when the hippocampus is reactivated by long-time retrieval of remote memory” at the end of 5^th^ paragraph of the Discussion.

“Reconsolidation is suggested to be a process not only to maintain memory, but also to enhance and/or update memory (Nader et al., 2000; Dudai, 2002; Tronel et al., 2005; Fukushima et al., 2014). […] Additionally, the possibility still remains that the activation (but not reactivation) of hippocampus by longer re-exposure to the familiar context is required to generate new memory associated with this familiar context.”

*3) There is a concern that rodent PTSD models based solely on fear conditioning do not distinguish between a physiological response to a stressful event and the pathophysiological alterations that may more closely reflects a PTSD-like state (Desmedt et al. al Biol Psych 2015). Furthermore, it appears that neurogenesis-promoting treatments would only be effective for hippocampal dependent memories (Figure 2). In this regard, PSTD memories are probably much more complex than those acquired during contextual-fear conditioning.*

Thank you for your critical comment. We absolutely agree with these comments, and now acknowledge the limitations of contextual fear as a model of PTSD. Specifically we now acknowledge that PTSD like state also should include other pathophysiological alterations of brain and other organs (and certainly goes much beyond the hippocampus). According to the suggestions from the reviewer, we added a sentence in the Discussion (last paragraph).

“In conclusion, we showed that forgetting of remote contextual fear memory was promoted by hippocampal neurogenesis following long (but not short) context reminders. […] Furthermore, it will be important to extend our findings to target hippocampus-independent traumatic memory since traumatic memory associated with PTSD must be more complex than contextual fear memory used in this study as the rodent PTSD model (Shin et al., 2006; Milad et al., 2007; Parsons et al., 2013; VanElzakker et al., 2014; Desmedt et al., 2015).”

*4) The treatments (memantine and voluntary exercise) do not completely abolish the fear memories (recent or remote), reducing conditioned freezing from ~50-60% to ~ 20-30% (the latter appears to be greater than the baseline/unconditioned freezing in most experiments). There should be some discussion of what this remaining memory represents, and whether it is due to remaining fear memory, sensitization, or non-hippocampal representations of fear.*

Thank you for your critical comments. As the reviewer pointed out, the treatments with neurogenesis enhancers failed to completely abolish the fear memories. As suggested, this may be due to remaining fear memory that depends on brain regions other than hippocampus. However, previous studies have shown that similar low levels of freezing responses are observed when consolidation or reconsolidation of fear memory is disrupted by amnestic drugs or gene manipulations and when fear memory is extinguished. Therefore, in this manuscript, we concluded that significant forgetting was observed following the treatments with MEM or running, but that the memory is not completely erased. Nonetheless, we found that c-fos expression is not induced in the ACC and hippocampus of mice received MEM treatments after the retrieval test compared to control groups (CS-US group without MEM treatments and no CS group, data not shown), suggesting that the forgetting observed is quite profound (above baseline levels of freezing, but no activation of ACC by retrieval at remote time point). According to the suggestions from reviewer, we added a paragraph to the Discussion.

“In this study, hippocampal neurogenesis enhancers (MEM and running) failed to completely abolish fear memories. […] In addition, we found that c-fos expression was not induced in the ACC and hippocampus when c-fos expression was assessed after the retrieval test following the treatments with MEM for 4 weeks (data not shown), strongly suggesting that the MEM treatment induces forgetting of fear memory even though mice still showed low level of freezing.”